# Neutrophil gelatinase-associated lipocalin partly reflects the dynamic changes of renal function among chronic hepatitis C patients receiving direct-acting antivirals

Yen-Chun Chen[1,2], Chen-Hao Li[1], Ping-Hung Ko[1], Chi-Che Lee[3], Ru-Jiang Syu[1], Chih-Wei Tseng[1,2☯]*, Kuo-Chih Tseng[1,2☯]*

1 Department of Internal Medicine, Dalin Tzu Chi Hospital, Buddhist Tzu Chi Medical Foundation, Chia-Yi, Taiwan, 2 School of Medicine, Tzu Chi University, Hualien, Taiwan, 3 Department of Medicine Research, Dalin Tzu Chi Hospital, Buddhist Tzu Chi Medical Foundation, Chia-Yi, Taiwan

☯ These authors contributed equally to this work.
* tsengkuochih@gmail.com (KCT); cwtseng2@gmail.com (CWT)

**Data Availability Statement:** As to sharing a de-identified data, data cannot be shared publicly because of privacy. Data are available from Ethics

## Abstract

### Background

Changes in renal function in chronic hepatitis C (CHC) patients receiving direct-acting antivirals (DAAs) are controversial. The evolution of neutrophil gelatinase-associated lipocalin (NGAL) in these patients remains unclear.

### Methods

A total of 232 CHC patients receiving DAA at Dalin Tzu Chi Hospital from May 2016 to February 2019, were enrolled in this retrospective study. Grade 2/3 renal function deterioration, defined as a decrease in eGFR between 10% and 50% from baseline (BL) to 12 weeks after the end of treatment (P12), was investigated for its association with BL characteristics. The changes in renal function and NGAL levels were also analyzed at the SOF-base or non-SOF-base DAA.

### Results

Sixty-two patients (26.7%) had grade 2/3 renal function deterioration at P12 after DAA therapy. Univariate analysis showed that it was associated with age (P = 0.038). Multivariate analysis indicated that age (OR = 1.033, 95% CI: 1.004–1.064, P = 0.027), sex (male; OR = 2.039, 95% CI: 1.093–3.804, P = 0.025), ACEI/ARB use (OR = 2.493, 95% CI: 1.016–6.119, P = 0.046), and BL NGAL (OR = 1.033, 95% CI: 1.001–1.067, P = 0.046) positively correlated with grade 2/3 renal function deterioration. Furthermore, eGFR was decreased (P = 0.009) and NGAL was increased (P = 0.004) from BL to P12 in CHC patients receiving SOF-based DAA.

Committee of Dalin Tzu Chi Hospital via E-mail: irb_DL@tzuchi.com.tw for researchers who meet the criteria for access to confidential data.

**Funding:** This study was funded by the Dalin Tzu Chi Hospital, Buddhist Tzu Chi Medical Foundation through grant numbers DTCRD108-I-10 and the Buddhist Tzui Chi Medical Foundation through grant number TCMF-MP 109-01-01. The sponsors played no role in the study design, collection, analysis, and interpretation of data; in the writing of the report; and in the decision to submit the article for publication.

**Competing interests:** The authors declare that they have no competing interests.

**Abbreviations:** HCV, hepatitis C virus; CKD, chronic kidney disease; ESRD, end-stage renal disease; DAA, direct-acting antivirals; CHC, chronic hepatitis C; eGFR, estimated glomerular filtration rate; SOF-based, sofosbuvir-based; NGAL, neutrophil gelatinase-associated lipocalin; AKI, acute kidney injury; EOT, end of treatment; P12, 12 weeks after end of treatment; BL, baseline; SVR, sustained virologic response; DAIDS, Division of AIDS; AST, aspartate aminotransferase; ALT, alanine aminotransferase; CKD-EPI, Chronic Kidney Disease Epidemiology Collaboration; FIB-4, fibrosis-4; DM, diabetes mellitus; HTN, hypertension; HCC, hepatocellular carcinoma; ELISA, enzyme-linked immunosorbent assay; SD, standard deviation; ACEI, angiotensin-converting enzyme inhibitor; ARB, angiotensin receptor blockers; and NSAID, nonsteroidal anti-inflammatory drug.

## Conclusions

Of the CHC patients receiving DAA therapy, 26.7% had grade 2/3 renal function deterioration at P12, and it was associated with older age, gender being male, ACEI/ARB use, and higher BL NGAL levels. In addition, NGAL might be a biomarker of nephrotoxicity at P12 in patients receiving SOF-based DAA.

## 1. Introduction

Chronic hepatitis C virus (HCV) infection may result in chronic kidney disease (CKD) and end-stage renal disease (ESRD) [1]. Currently, interferon-free direct-acting antivirals (DAAs) are the standard therapy for chronic hepatitis C (CHC) [2–4] and can improve some HCV-related extrahepatic outcomes [1]. However, the short-term effect of DAA on renal function is inconclusive [5,6]. Several studies have reported that the estimated glomerular filtration rate (eGFR) might be reduced during DAA therapy, especially sofosbuvir (SOF) -based regimens [7–13]. However, some studies have revealed that the eGFR change might be nonsignificant [10,11,13–16]. The metabolite of SOF is mainly cleared from the body via the kidney, but other kinds of DAAs are primarily metabolized and cleared by the liver [1], SOF-based DAA could be nephrotoxic.

The biomarker neutrophil gelatinase-associated lipocalin (NGAL) is a protein involved in complex biological activities such as bacteriostatic effects, cell proliferation and differentiation, and cellular apoptosis. The levels would markedly increase if renal tubular injury occurred [17]. NGAL has been found to be a useful tool to detect acute kidney injury (AKI) earlier than the traditional method estimated by creatinine in patients with drug-induced nephrotoxicity or those undergoing cardiac surgery and liver transplantation. It is also a predictor of delayed graft function after kidney transplantation [17–19]. In addition, the performance of urine or plasma/serum NGAL levels was comparable [19]. However, there is a paucity of studies investigating the relationship between renal function and NGAL in CHC patients treated with DAA, and the results are conflicting [20–22]. Strazzulla et al reported that NGAL levels increased at week 12 after DAA therapy, but the earlier generation of DAAs such as telaprevir was included in that study [20]. Regarding the relationship between NGAL and SOF-based DAA, some results are conflicting. Strazzulla et al. reported that NGAL levels would increase at the end of treatment (EOT) of DAA and at 12 weeks after EOT (P12) when compared with baseline (BL); however, Ali Nada et al. stated that NGAL would be decreased at EOT when compared with BL [21,22]. Furthermore, the relationship between nonSOF-based DAA, renal function, and NGAL is unknown; therefore, we conducted this study to investigate the changes in eGFR and the role of NGAL in CHC patients receiving DAA, including nonSOF- and SOF-based regimens.

## 2. Patients and methods

### 2.1 Patient selection

CHC patients who underwent DAA treatment between May 2016 and February 2019 from the Dalin Tzu Chi hospital were enrolled in this retrospective study. All patients were positive for anti-hepatitis C antibody for more than 6 months and had detectable serum HCV RNA at the time of enrollment. The treatment duration and regimens were based on the guidelines [3,4]. Patients with eGFR less than 30 ml/min/1.73 m$^2$, active cancer status, hepatitis B virus co-

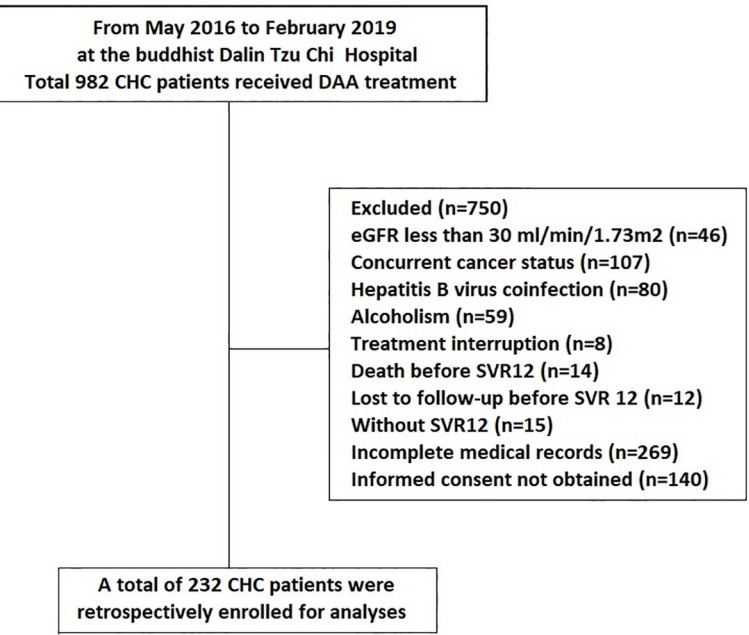

Fig 1. Flowchart of patient selection.

infection, alcoholism, incomplete treatment course, incomplete medical records, died during the treatment course, missed post-treatment follow-up, or without sustained virologic response 12 (SVR12) were excluded. A total of 982 patients were screened, and 232 patients were enrolled in this study (Fig 1). The study protocol conformed to the ethical guidelines of the 1975 Declaration of Helsinki as reflected in a priori approval by the Ethics Committee of Dalin Tzu Chi Hospital (B10502022, B10704016). Written informed consent was obtained from all patients during enrollment.

## 2.2 Clinical and laboratory monitoring

According to the Division of AIDS (DAIDS) Table for Grading the Severity of Adult and Pediatric Adverse Events, grade 2 and 3 (grade 2/3) renal function deterioration were defined as a decrease of eGFR $\geq$ 10% but < 30% and $\geq$ 30% but < 50%, respectively, in our population [23]. Grade 4 renal function deterioration was defined as a $\geq$ 50% decrease from participant's BL. The primary outcomes were BL factors associated with grade 2/3 and grade 4 renal function deterioration at P12 after DAA treatment. The secondary outcomes were the evolution of eGFR and NGAL in the different subgroups.

Laboratory assessments (serum aspartate aminotransferase [AST], alanine aminotransferase [ALT], albumin, total bilirubin, hemoglobin, prothrombin time) and abdominal ultrasonography were performed at the BL, EOT, and P12. HCV RNA was quantified at the BL, EOT, and P12. The eGFR calculated using the Chronic Kidney Disease Epidemiology Collaboration (CKD-EPI) equation [24] and NGAL were both obtained at the BL, EOT, and P12. According to the BL eGFR, patients' renal function was classified as rank 1 (eGFR: > 90 ml/min/1.73 m$^2$), rank 2 (eGFR: 60–90 ml/min/1.73 m$^2$), and rank 3 (eGFR: 30–60 ml/min/1.73 m$^2$). Fibrosis-4 (FIB-4) was used as a noninvasive test for liver fibrosis [25,26]. Advanced liver fibrosis was diagnosed by an FIB-4 score higher than 3.25 [25,26] or radiologic cirrhosis [27]. Radiologic cirrhosis was defined as coarse liver echotexture

with nodularity and small liver size or the presence of features of portal hypertension (e.g., splenomegaly, ascites, or varices) noted on imaging. A diagnosis of fatty liver was based on findings from abdominal ultrasound, including the features of hepatorenal echogenicity contrast, liver brightness, deep attenuation, and vessel blurring [28]. Other clinical factors, including chronic hepatitis B, hyperlipidemia, diabetes mellitus (DM), hypertension (HTN), IFN-experienced, co-medications, hepatocellular carcinoma (HCC), and alcoholism, were recorded by chart review. Alcoholism was defined as alcohol consumption of more than 40 g/day [29]. HCC was diagnosed either by biopsy or by imaging in the setting of liver cirrhosis [30].

### 2.3 HCV quantification and genotyping

Serum HCV RNA was quantified at the BL, EOT, and P12 using the COBAS AmpliPrep/COBAS TaqMan HCV Test, v2.0 (Roche Diagnostics, Rotkreuz, Switzerland), with a lower limit of quantification of 15 IU/mL. HCV genotyping was performed using the COBAS HCV GT (Roche Diagnostics).

### 2.4 Neutrophil gelatinase-associated lipocalin (NGAL) assay

NGAL was measured in serum samples using a sandwich enzyme-linked immunosorbent assay (ELISA; FineTest, Wuhan, China), following the manufacturer's recommendations [31]. Samples were stored at −40˚C until analysis. The lower detection limit of the NGAL assay was 0.19 ng/mL. The normal range is 0.313–20 ng/mL. Quality control samples were also included in each assay.

### 2.5 Statistical analysis

The commercial statistical software package (SPSS for Windows, version 22) was used for all statistical analyses. Variables were expressed as frequency count, percentage of total, and mean ± standard deviation. Basic comparisons of demographics and baseline clinical features between the groups with and without grade 2/3 renal function deterioration at P12 were firstly performed by univariate analyses by logistic regressions. In addition, multivariate logistic regression analysis was performed to evaluate the factors associated with renal function deterioration after DAA treatment, including those with P value < 0.1, after univariate analysis, and comorbidities [32,33] potentially making kidneys susceptible to injury. All statistical tests were two-tailed, with P < 0.05 considered as being significant.

## 3. Results

### 3.1 Baseline characteristics of CHC patients

A total of 232 patients were included in the analysis. The BL characteristics are listed in Table 1. This cohort included 83 men (35.8%) with a mean age of 64.0±10.7 years. Most patients were infected with HCV genotype 1 (n = 154, 66.4%). Of the patients in the cohort, 76 had BL eGFR > 90 ml/min/1.73 m$^2$ (rank 1, 32.8%), 120 had BL eGFR between 60 and 90 ml/min/1.73 m$^2$ (rank 2, 51.7%), and 36 had BL eGFR between 30 and 60 ml/min/1.73 m$^2$ (rank 3, 15.5%). Forty-four patients had DM (19.0%); 50 had HTN (21.6%). Twenty-five patients were angiotensin-converting enzyme inhibitor (ACEI)/angiotensin receptor blocker (ARB) users (10.8%), 11 were diuretic users (4.7%), and 34 were nonsteroidal anti-inflammatory drug (NSAID) users (14.7%). Of these, 153 patients had advanced fibrosis (65.9%) and 37 had a history of HCC (15.9%). Paritaprevir/ritonavir/ombitasvir with dasabuvir (n = 71, 30.6%), sofosbuvir/ledipasvir (n = 55, 23.7%),

**Table 1. Baseline characteristics of chronic hepatitis C patients receiving DAA with or without grade 2/3 renal function deterioration at P12.**

| Variable | All patients (n = 232) | With grade 2/3 deterioration (n = 62) (26.7%) | Without grade 2/3 deterioration (n = 170) (73.3%) | P-value[#] |
|---|---|---|---|---|
| **Baseline clinical characteristics** | | | | |
| Age (years)† | 64.02 ± 10.65 | 66.44 ± 8.49 | 63.14 ± 11.23 | 0.038 |
| Male (%) | 83 (35.8%) | 28 (45.2%) | 55 (32.4%) | 0.073 |
| Fatty liver | 75 (32.3%) | 23 (37.1%) | 52 (30.6%) | 0.349 |
| Hyperlipidemia | 18 (7.8%) | 8 (12.9%) | 10 (5.9%) | 0.084 |
| Diabetes mellitus | 44 (19.0%) | 14 (22.6%) | 30 (17.6%) | 0.397 |
| Hypertension | 50 (21.6%) | 16 (25.8%) | 34 (20.0%) | 0.342 |
| eGFR ranks* | | | | 0.211 |
| rank 1 | 76 (32.8%) | 17 (27.4%) | 59 (34.7%) | |
| rank 2 | 120 (51.7%) | 33 (53.2%) | 87 (51.2%) | |
| rank 3 | 36 (15.5%) | 12 (19.4%) | 24 (14.1%) | |
| **Baseline characteristics of HCV and liver-related conditions** | | | | |
| Advanced fibrosis (%)‡ | 153 (65.9%) | 45 (72.6%) | 108 (63.5%) | 0.200 |
| HCC history (%) | 37 (15.9%) | 14 (22.6%) | 23 (13.5%) | 0.099 |
| Splenomegaly (%) | 71 (30.6%) | 25 (40.3%) | 46 (27.1%) | 0.054 |
| Ascites (%) | 5 (2.2%) | 2 (3.2%) | 3 (1.8%) | 0.504 |
| Baseline HCV viral load (IU/mL)† | 5.99Log ± 0.97Log | 5.87Log ± 1.14Log | 6.04Log ± 0.91Log | 0.238 |
| HCV genotype 1 (%) | 154 (66.4%) | 43 (69.4%) | 111 (65.3%) | 0.563 |
| Sofosbuvir-based (%) | 112 (48.3%) | 33 (53.2%) | 79 (46.5%) | 0.363 |
| **Baseline medications associated with renal function** | | | | |
| ACEI/ARB users | 25 (10.8%) | 11 (17.7%) | 14 (8.2%) | 0.043 |
| Diuretics users | 11 (4.7%) | 4 (6.5%) | 7 (4.1%) | 0.463 |
| NSAID users | 34 (14.7%) | 7 (11.3%) | 27 (15.9%) | 0.384 |
| **Baseline laboratory data** | | | | |
| Baseline NGAL (ng/ml) † | 16.10 ± 9.03 | 18.00 ± 10.00 | 15.40 ± 8.58 | 0.055 |
| ALT (U/L)† | 86.60 ± 75.20 | 93.08 ± 81.93 | 84.24 ± 72.69 | 0.430 |
| AST (U/L)† | 61.70 ± 55.22 | 69.68 ± 67.20 | 58.79 ± 50.07 | 0.194 |
| Albumin (g/dl) † | 4.20 ± 0.37 | 4.14 ± 0.39 | 4.22 ± 0.35 | 0.137 |
| Total bilirubin (mg/dl) † | 0.79 ± 0.40 | 0.83 ± 0.39 | 0.77 ± 0.41 | 0.349 |
| eGFR (ml/min/1.73m$^2$)† | 78.94 ± 17.82 | 76.43 ± 17.04 | 79.86 ± 18.05 | 0.195 |
| Hb (gm/dL) † | 13.50 ± 1.62 | 13.48 ± 1.53 | 13.51 ± 1.66 | 0.916 |
| Prothrombin time (INR)† | 1.03 ± 0.07 | 1.04 ± 0.07 | 1.03 ± 0.07 | 0.094 |

e-GFR, estimated glomerular filtration rate; HCC, hepatocellular carcinoma; HCV, hepatitis C virus; ACEI, angiotensin-converting enzyme inhibitor; ARB, angiotensin receptor blocker; NSAID, nonsteroidal anti-inflammatory drugs; NGAL, neutrophil gelatinase-associated lipocalin; ALT, alanine aminotransferase; AST, aspartate aminotransferase; INR, international normalized ratio.

*rank 1: > 90 ml/min/1.73 m2, rank 2: 60–90 ml/min/1.73 m2, rank 3: 30–60 ml/min/1.73 m2

†Data are expressed as mean±SD

#the P-value here was calculated by univariate logistic regression analysis

‡Advanced liver fibrosis was diagnosed by an FIB-4 ≧ 3.25 or radiologic cirrhosis. Radiologic cirrhosis was defined as coarse liver echotexture with nodularity and small liver size or the presence of features of portal hypertension (e.g., splenomegaly, ascites, or varices) noted on imaging.

sofosbuvir with ribavirin (n = 45, 19.4%), elbasvir/grazoprevir (n = 31, 13.4%), Glecaprevir/ Pibrentasvir (n = 18, 7.8%) and sofosbuvir/daclatasvir (n = 12, 5.2%) were the DAAs used in our studied population.

## 3.2 Baseline factors associated with renal function deterioration

The BL characteristics of patients with and without grade 2/3 renal function deterioration are shown in Table 1. None of the patients had grade 4 renal function deterioration according to the DAIDS table. Sixty-two patients (26.7%) had grade 2/3 renal function deterioration at P12 after DAA therapy, and such deterioration was only associated with age (P = 0.028) on univariate analysis. Other factors, including hyperlipidemia, DM, HTN, different BL eGFR ranks, liver function tests, BL HCV viral load, previous HCC history, and SOF-based DAA regimens were not significantly different between the two groups. The percentage of users of commonly used medications that may affect eGFR, such as ACEI, ARB, diuretics, and NSAIDs, was not significantly different between those with or without grade 2/3 renal function deterioration on univariate analysis. In multivariate analysis, age (OR = 1.033, 95% CI: 1.004–1.064, P = 0.028), gender (male) (OR = 2.021, 95% CI: 1.083–3.771, P = 0.027), ACEI/ARB use (OR = 2.476, 95% CI: 1.009–6.075, P = 0.048), and BL NGAL (OR = 1.033, 95% CI: 1.001–1.067, P = 0.045) were positively correlated with grade 2/3 renal function deterioration (Table 2).

## 3.3 Serial changes of eGFR and NGAL in overall patients, patients receiving nonSOF- or SOF-based DAA regimens, and different BL eGFR ranks

The serial changes in eGFR from the BL to P12 in all patients are shown in Fig 2A. The mean eGFR was 78.94±17.82 ml/min/1.73 m$^2$ at the BL and decreased to 76.41±18.25 ml/min/1.73 m$^2$ and 77.32±18.48 ml/min/1.73 m$^2$ at EOT and P12, respectively (EOT vs. BL, P < 0.001; P12 vs. BL, P = 0.005). The overall NGAL at the BL, EOT and P12 (Fig 2B) were 16.10±9.03 ng/ml, 16.13±8.54 ng/ml and 16.50±8.60 ng/ml, respectively (BL vs EOT, P = 0.274; BL vs P12, P = 0.056; EOT vs P12, P = 0.159).

The eGFRs in patients receiving nonSOF-based DAA were significantly decreased from the BL to EOT (BL vs. EOT, P = 0.007). However, the eGFR did not significantly decrease from the BL to P12 (BL vs. P12, P = 0.173) (Fig 3A). The eGFRs in patients receiving SOF-based DAA were significantly decreased from the BL to EOT and from the BL to P12 (BL vs. EOT, P = 0.001; BL vs. P12, P = 0.009) (Fig 3A). The eGFR at the BL, EOT, and P12 were not significantly different between the non-SOF-based and SOF-based groups (P = 0.131, 0.141, and 0.077, respectively). For nonSOF-based DAA users, the NGAL levels at the BL, EOT, and P12 were not significantly different from each other (BL vs. EOT, P = 0.986; BL vs. P12, P = 0.937; EOT vs. P12, P = 0.428) (Fig 3B). For SOF-based DAA users, the NGAL levels at the BL, EOT,

**Table 2. Factors associated with grade 2/3 renal function deterioration in chronic hepatitis C patients receiving DAA at P12^.**

|  | Odds Ratio (95% CI) | P |
|---|---|---|
| Age | 1.033 (1.004–1.064) | 0.027 |
| Sex | | |
| Female | 1.000 | 0.025 |
| Male | 2.039 (1.093–3.804) | |
| ACEI/ARB | | |
| Non-user | 1.000 | 0.046 |
| User | 2.493 (1.016–6.119) | |
| BL NGAL | 1.033 (1.001–1.067) | 0.046 |

^Adjusted for age, sex, variables with P < 0.1 from Table 1: hyperlipidemia, splenomegaly, HCC history, ACEI/ARB users, BL NGAL and PT INR; factors reported to be associated with renal injury: baseline renal disease, DM, HTN, liver disease (advanced fibrosis), diuretics users, NSAID users, and SOF users.

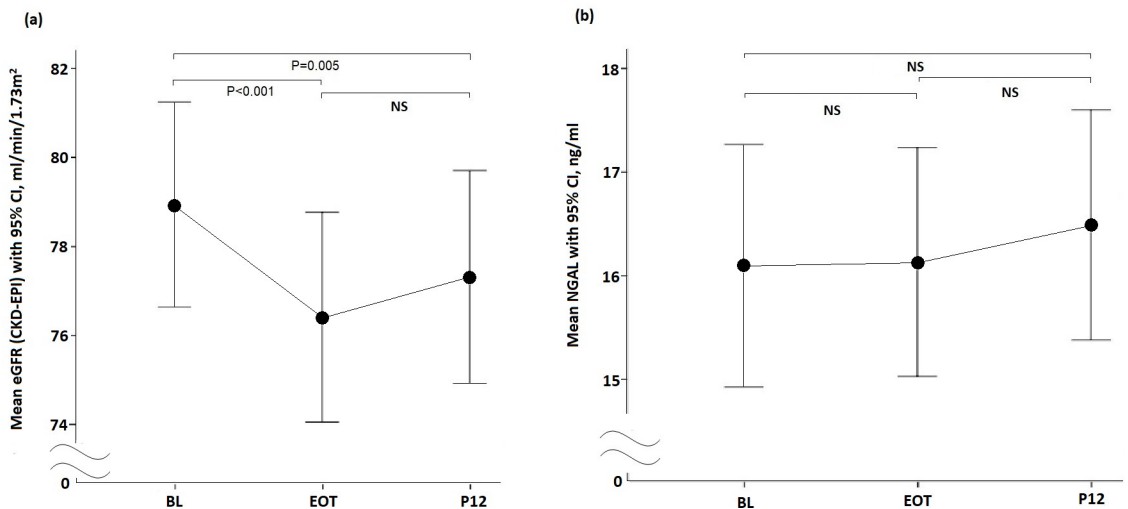

**Fig 2. Overall eGFR and NGAL changes from the BL, EOT to P12 in CHC patients receiving DAA therapy.** a. eGFR; b. NGAL.

and P12 were mildly increased, with a significant difference between the BL and P12 (BL vs. EOT, P = 0.112; BL vs. P12, P = 0.004; EOT vs. P12, P = 0.236) (Fig 3B). The levels of NGAL at the BL, EOT, and P12 were all significantly different between nonSOF- and SOF-based DAA users (P = 0.011, 0.024, and 0.037, respectively) (Fig 3B). Further subgroup analysis showed predictive factors associated with grade 2/3 renal function deterioration were different between nonSOF- and SOF-based users (n = 120 and n = 112, respectively) (S1–S3 Tables). The predictive factors of nonSOF users (n = 120) were gender (OR:3.161; CI: 1.150–8.686; P = 0.026), fatty liver (OR: 4.684; CI: 1.689–12.990; P = 0.003), and hyperlipidemia (OR:9.401; CI: 2.268–38.959; P = 0.002). The BL NGAL has the trend associated with the grade 2/3 renal function deterioration (P = 0.05). The ACEI/ARB use (OR:3.276; CI: 1.008–10.647; P = 0.049) was the only predictive factor for SOF users.

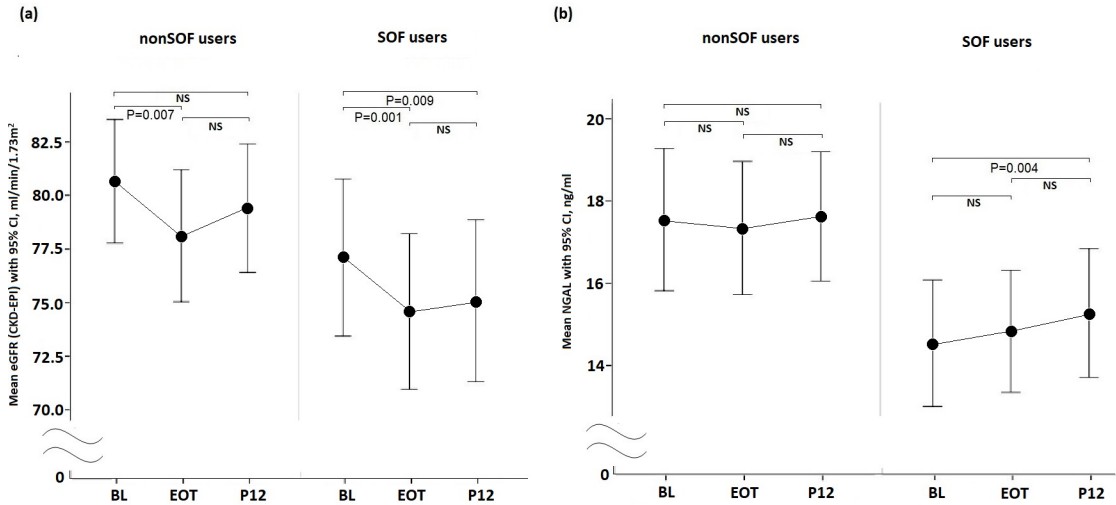

**Fig 3. The eGFR and NGAL changes from the BL, EOT to P12 in nonSOF-based or SOF-based DAA subgroups.** a. eGFR; b. NGAL.

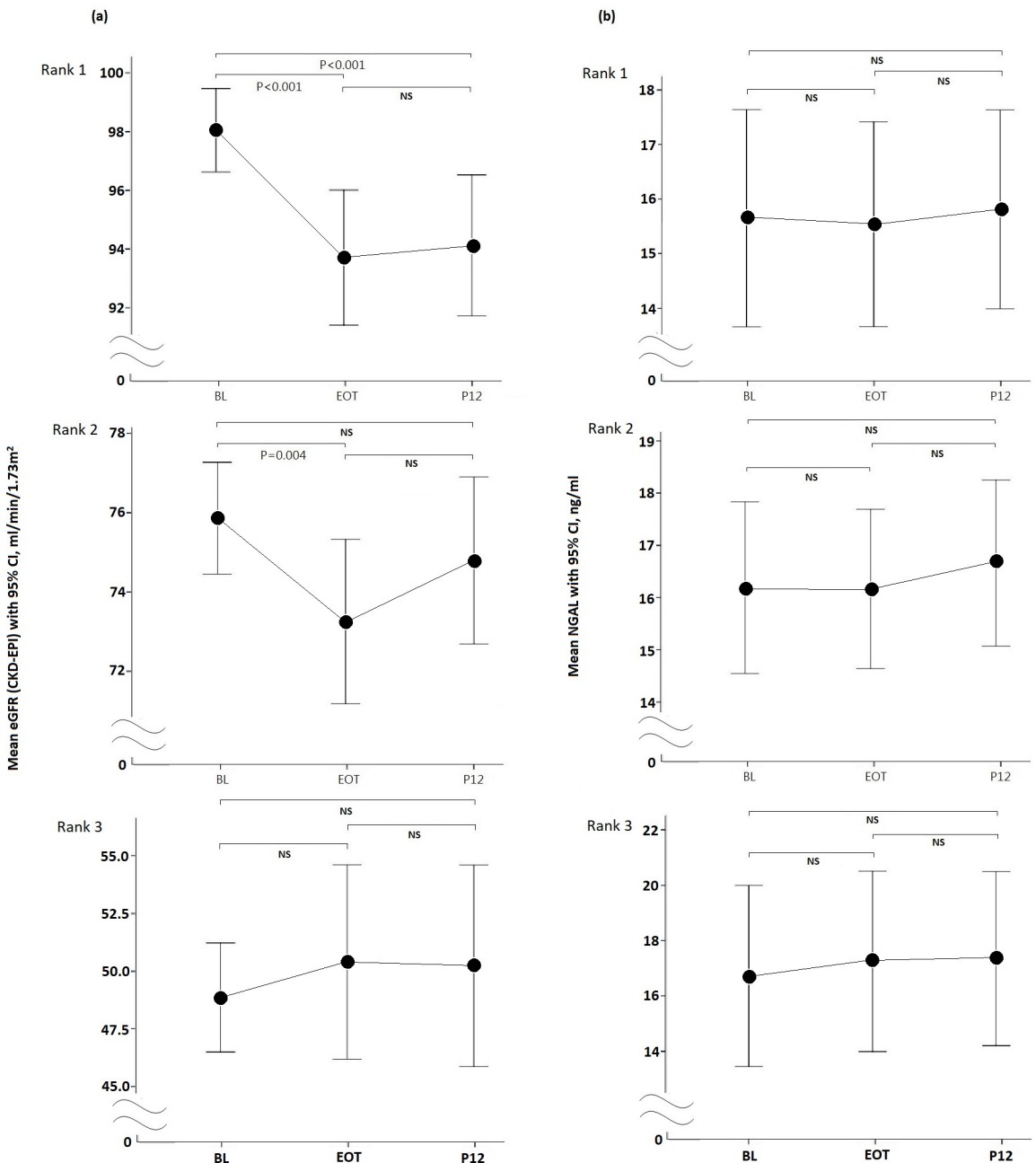

**Fig 4. The eGFR and NGAL changes from the BL, EOT to P12 in different BL eGFR ranks in CHC patients receiving DAA therapy.** a. eGFR; b. NGAL.

As shown in Fig 4A, among patients with BL eGFR rank 1, the eGFR was significantly reduced from the BL to EOT and from the BL to P12, but not from EOT to P12 (BL vs. EOT, P < 0.001; BL vs. P12, P < 0.001; EOT vs. P12, P = 0.680). Among patients with BL eGFR rank 2, the eGFR was significantly decreased from the BL to EOT but not significantly different between BL and P12 and between EOT and P12 (BL vs. EOT, P = 0.004; BL vs. P12, P = 0.225; EOT vs. P12, P = 0.086). Among patients with BL eGFR rank 3, the eGFR was not significantly different between the two groups (BL vs. EOT, P = 0.423; BL vs. P12, P = 0.477; EOT vs. P12,

P = 0.910). As shown in Fig 4B, among patients with BL eGFR rank 1, 2, or 3, the NGAL levels at the BL, EOT, and P12 were not significantly different from each other (for rank 1: BL vs. EOT, P = 0.755; BL vs. P12, P = 0.626; EOT vs. P12, P = 0.396; for rank 2: BL vs. EOT, P = 0.935; BL vs. P12, P = 0.152; EOT vs. P12, P = 0.105; for rank 3: BL vs EOT, P = 0.313; BL vs P12, P = 0.337; EOT vs. P12, P = 0.864). The levels of NGAL at the BL, EOT, and P12 were not significantly different between BL eGFR ranks 1, 2, and 3 (P = 0.831, 0.613, and 0.646, respectively).

## 4. Discussion

This study showed that older patients, males, ACEI/ARB users, and those with higher BL NGAL levels were associated with grade 2/3 renal function deterioration at P12 (Table 2). Moreover, the serial changes in eGFR from both nonSOF- and SOF-based DAA users were significantly decreased from the BL to EOT, and this decrease was only persistent at P12 in SOF-based users. The serum levels of NGAL were significantly increased at P12 from the BL for SOF-based DAA users but were similar among the BL, EOT, and P12 for non-SOF-based DAA users.

The overall eGFR (Fig 2A) of DAA-treated CHC patients decreased from the BL to EOT and P12 after DAA treatment. Several studies have revealed similar phenomena [9,11,13]. However, a decreased eGFR may not reflect true renal injury. A previous study reported that the first-generation DAA, telaprevir, was found to affect creatinine transporters, and therefore, the values of eGFR [34]. Hence, the level of creatinine and eGFR would not fully show the true renal function in such circumstances. The overall levels of NGAL (Fig 2B) did not change significantly during this period when eGFR was reduced. This result could only show that it is less likely to have tubular injury, but other possibilities such as glomerular damage could not be ruled out.

Grade 2/3 renal function deterioration was observed in 26.7% of patients at P12. Multivariate analyses (Table 2) revealed that older age, gender (male), ACEI/ARB use, and higher levels of BL NGAL were independent risk factors. Older age has also been reported as a factor for renal function deterioration after short-term follow-up in patients with DAA-treated CHC patients [11,13]. Patients older than 60 years are at a highest risk of drug-induced nephrotoxicity [35]. In this study, we also showed a positive association between male sex and renal function deterioration after DAA treatment, which has not been reported in other studies. The most likely reasons could be that the definition of renal injury, the follow-up period, and the confounding factors for multivariate analysis were different in these studies. In fact, there are conflicting reports about the influence of gender on the development of AKI [35]. ACEI/ARB users may have decreased eGFR due to the adverse effects of these medications, which might be exacerbated by drug-drug interactions from DAA with ACEI/ARB. Patients with any of these risk factors, especially those who have more than one risk factor (e.g., an old male patient), should be closely monitored for changes in renal function when a nephrotoxic medication is added or the dosage is increased. To our knowledge, this is the first report to describe the positive association between BL NGAL levels and renal function deterioration at P12 during DAA therapy. The exact mechanism was unclear in our study. It is also difficult to hypothesize due to the complex activities of NGAL, which are related to antimicrobial effects, cell differentiation, acute-phase response, and renal tubular injury [36]. The role of NGAL in the early detection of drug-induced nephrotoxicity (such as cisplatin, aminoglycosides, or amphotericin B) has been investigated in animal and human studies. However, more human studies involving urine or serum NGAL levels, are necessary [18].

Since SOF might be related to renal injury via drug-induced tubulointerstitial nephritis [12], we further subdivided our population into nonSOF-based and SOF-based DAA users for comparison. As shown in Fig 3A, the eGFRs at EOT were significantly decreased in both subgroups (nonSOF-based, P = 0.007; SOF-based, P = 0.001) when compared with the BL. This eGFR decrease at P12 was still significant for SOF-based users (P = 0.009) but became nonsignificant for nonSOF-based users when compared with the BL. This phenomenon might indirectly show nephrotoxicity from sofosbuvir during treatment, and this decrease persisted up to P12. Another study also demonstrated that the SOF-related eGFR decline was more prominent during the on-treatment period, but the eGFR decline improved after the off-treatment period [12]. The study indicated that the renal toxicity of sofosbuvir may be reversible after long-term follow-up. We further found that NGAL was significantly increased at P12 (P = 0.004) when compared with the BL in SOF-based DAA users, but the levels of NGAL between the BL, EOT, and P12 showed no significant changes in nonSOF-based DAA users. Our results are similar to those of a previous study by Strazzulla et al. [21] Strazzulla et al. showed that NGAL was significantly increased at P12 when compared with the BL, but the eGFR was not. However, another study by Ali Nada et al. reported that NGAL levels decreased after SOF-based therapy at EOT [22]. These conflicting results may be due to differences in ethnicity (Taiwanese in our study; Italian and Egyptian in the other two studies, respectively), or different HCV genotypes (mainly genotype 1b in our and Strazzulla et al study, but mainly genotype 4 in Ali Nada et al study). In addition, Ali Nada et al. did not check NGAL at P12; therefore, we could not directly compare these studies. Although we found that an increase in NGAL was significant at P12 by SOF-based DAA, the values were still within the normal range (normal range is 0.313–20 ng/ml). Finally, we performed univariate and multivariate analyses again for nonSOF and SOF-based DAA users separately (S1–S3 Tables). Interestingly, the predictive factors of grade 2/3 renal function deterioration among these two subgroups were different. The possible mechanisms are unclear. Hence, further prospective studies with larger populations and different HCV genotypes are required to clarify the evolutions of renal function and NGAL during non-SOF- or SOF-based DAA therapy.

Many studies have shown that a lower BL eGFR was associated with eGFR improvement post-DAA treatment [9,11,13,14,16,37]. Therefore, we stratified patients with BL eGFR into ranks 1, 2, and 3 (Table 1) and assessed the changes in eGFR at the BL, EOT, and P12 (Fig 4A). The changes in eGFR from the BL to EOT and P12 in different BL eGFR subgroups demonstrated that eGFR would be significantly decreased at EOT and P12 in those with higher BL eGFR (rank 1) but significantly decreased only at EOT in those with BL eGFR between 60 and 90 ml/min/1.73 $m^2$ (rank 2). These findings have also been reported in other studies [11,13]. In those studies, a better BL eGFR was associated with a reduced eGFR at P12. One study even found that DAA treatment for CHC patients would be helpful in improving eGFR decline only among those with baseline eGFR less than 60 ml/min/1.73 $m^2$ [16]. The authors thought that some CHC patients with HCV-induced glomerular disease are more likely to have renal function improvement after HCV eradication by DAA. Another study [9] reported that more patients experienced a decrease in eGFR by >10 ml/min/1.73 $m^2$ in those with BL eGFR > 60 ml/min/1.73 $m^2$. Taken together, it seems that patients with a higher BL eGFR are more likely to have decreased eGFR after DAA therapy and the exact mechanism remains unclear. However, BL eGFR was not an independent predictor of renal function deterioration after multivariate analysis in our study. Moreover, NGAL was not correlated with eGFR changes in the different BL eGFR subgroups.

There are some limitations in our study. First, the choice of the DAA regimen was not randomized. Therefore, selection bias may exist. For example, some physicians would choose SOF-based DAA for patients with more advanced liver disease or a history of decompensation.

However, we know that protease inhibitors containing DAAs are not suitable for decompensated cirrhosis. Therefore, randomization is not ethical. Second, we did not check other biomarkers of acute renal damage, such as cystatin C or kidney injury molecule 1, which may be helpful in combination with NGAL as NGAL is mainly a biomarker for detecting tubular damage. However, renal injury may occur in the glomeruli, proximal and distal tubules, or loops of Henle [38,39]. NGAL is also produced by immune cells and the liver [36]. Therefore, once the injury caused by DAAs is mild, or when we consider other sources of NGAL, the role of NGAL might not be prominent. Third, we did not check these patients' urinalyses or their albuminuria levels, which is also suggestive of renal injury. Fourth, renal biopsy was not performed during or after DAA therapy. The histological findings may provide us with more information about DAA-related kidney injury and the possible underlying mechanisms. However, renal biopsy is risky, especially when only a relative minority of patients experience more than grade 2 renal function deterioration, and most deteriorations are reversible after long-term follow-up [12]. Hence, it is less meaningful to perform a renal biopsy under such circumstances. In addition, regardless of the adverse renal effect of DAA, we should closely follow up renal function in CHC patients with CKD even after HCV eradication by DAA therapy because of their fragile renal function.

## 5. Conclusion

A total of 26.7% of CHC patients receiving DAA therapy had a decreased eGFR of more than 10% during short-term follow-up, especially in older patients, males, ACEI/ARB users, and those with higher BL NGAL levels. In addition, NGAL might be a biomarker of nephrotoxicity at P12 in patients receiving SOF-based DAA.

## Supporting information

**S1 Table. Baseline characteristics of chronic hepatitis C patients receiving DAA with or without grade 2/3 renal function deterioration at P12 in nonSOF users.**
(DOCX)

**S2 Table. Baseline characteristics of chronic hepatitis C patients receiving DAA with or without grade 2/3 renal function deterioration at P12 in SOF users.**
(DOCX)

**S3 Table. Patient factors associated with grade 2/3 renal function deterioration in chronic hepatitis C patients receiving DAA at P12^ for nonSOF and SOF-based DAA users.**
(DOCX)

## Acknowledgments

We would like to thank Miss Yi-Ting He for her assistance in recording the records.

## Author Contributions

**Conceptualization:** Yen-Chun Chen, Ping-Hung Ko, Ru-Jiang Syu, Kuo-Chih Tseng.

**Data curation:** Ping-Hung Ko, Chi-Che Lee, Ru-Jiang Syu.

**Formal analysis:** Yen-Chun Chen, Chih-Wei Tseng, Kuo-Chih Tseng.

**Investigation:** Yen-Chun Chen, Chi-Che Lee, Chih-Wei Tseng, Kuo-Chih Tseng.

**Methodology:** Chen-Hao Li, Chih-Wei Tseng, Kuo-Chih Tseng.

**Resources:** Ping-Hung Ko.

**Writing – original draft:** Yen-Chun Chen.

**Writing – review & editing:** Chen-Hao Li, Chih-Wei Tseng, Kuo-Chih Tseng.

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
