## [Decision Letter · Decision Letter 0]

1 Jun 2021

PONE-D-21-15486

Neutrophil gelatinase-associated lipocalin partly reflects the dynamic changes of renal function among chronic hepatitis C patients receiving direct-acting antivirals

PLOS ONE

Dear Dr. Chen,

Thank you for submitting your manuscript to PLOS ONE. After careful consideration, we feel that it has merit but does not fully meet PLOS ONE’s publication criteria as it currently stands. Therefore, we invite you to submit a revised version of the manuscript that addresses the points raised during the review process.

We look forward to receiving your revised manuscript.

Kind regards,

Chen-Hua Liu

Academic Editor

PLOS ONE

Reviewers' comments:

Reviewer's Responses to Questions

**Comments to the Author**

1. Is the manuscript technically sound, and do the data support the conclusions?

Reviewer #1: Yes

Reviewer #2: Yes

2. Has the statistical analysis been performed appropriately and rigorously? 

Reviewer #1: Yes

Reviewer #2: Yes

3. Have the authors made all data underlying the findings in their manuscript fully available?

Reviewer #1: Yes

Reviewer #2: Yes

4. Is the manuscript presented in an intelligible fashion and written in standard English?

Reviewer #1: Yes

Reviewer #2: Yes

5. Review Comments to the Author

Reviewer #1: Authors proposed neutrophil gelatinase-associated lipocalin partly reflects the dynamic changes of renal function among CHC patients receiving DAA. It would give another point of view for clinical daily practice. I have several questions as below.

1. In table 1, authors showed baseline characteristics of subgroup with or without grade 2/3 renal function deterioration at P12. I was confused that the P-value here (in this table) was calculated by the Student's t-test and the Mann-Whitney U test for continuous variables/the chi-square test or Fisher's exact test for categorical data to check baseline difference between subgroups; "Or" by univariate logistic regression analysis? I suggest that authors should show another table for data of univariate logistic regression analysis for each predictive factors and then recheck multivariate logistic regression analysis.

2. In table 1, because many patients usually has abnormal liver function before treatment, FIB-4 might be influenced by hepatitis. I suggest "advanced liver fibrosis" might be replaced by "FIB-4 score higher than 3.25".

3. In table 1, HCC history means curative or inactive or stable status of HCC. Maybe authors exclude these patients to avoid some confounding effects.

4. In table 2, the odds ratio (OR) of ACEI/ARB was 2.476 (P=0.048) and the OR of BL NGAL was just 1.033 (P=0.045). ACEI/ARB had the higher weight for grade 2/3 renal function deterioration. If possible, authors could exclude patients using the medications associated with renal function (in table 1, ACEI/ARB, diuretics, NSAID) to clarify the effect of BL NGAL.

5. In Figure 3 & 4, the dynamic change of NGAL might be negatively associated with the dynamic change of eGFR from BL to P12, for patients with eGFR rank 1 and 2. This phenomenon was also found while focusing on SOF user. Therefore, these findings demonstrated the dynamic change of NGAL might reflect the dynamic changes of renal function. Maybe, authors could add the dynamic change of NGAL as one of predictive factors for analysis. As for BL NGAL, it should recheck by univariate/multivariate logistic regression analysis.

Reviewer #2: This retrospective study was aimed to investigate the changes in eGFR and the role of neutrophil gelatinase-associated lipocalin (NGAL) in CHC patients receiving DAA, including nonSOF- and SOF-based regimens. Although it is interesting, some concerns are needed to be clarified.

1. Since SOF might be related to renal injury via drug-induced tubulointerstitial nephritis, the proportion of SOF-based DAA users should be added in Table 1.

2. Factors associated with grade 2/3 renal function deterioration in CHC patients receiving DAA at P12 in nonSOF-based and SOF-based DAA users may be analyzed in another two tables, respectively. The cut -off of variables, including age, BL NGAL, should be offered in Table 2.

3. Page 16, line 3, “The levels of NGAL at the BL, EOT, and P12 were all significantly different between nonSOF- and SOF-based DAA users (P = 0.011, 0.024, and 0.037, respectively). Please add a figure.

4. Page 21, line 6, “Although we found that an increase in NGAL was significant at P12 by SOF-based DAA, the values were still within the normal range”. What is the normal range of NGAL?

5. Page 22, “Fig 4. The eGFR and NGAL changes from the BL, EOT to P12 in different BL eGFR ranks in CHC patients receiving DAA therapy. a. eGFR; b. NGAL” should be in “3. Results”.

6. PLOS authors have the option to publish the peer review history of their article (what does this mean?). If published, this will include your full peer review and any attached files.

Reviewer #1: No

Reviewer #2: No

---

## [Author Response · Author response to Decision Letter 0]

20 Jul 2021

Reviewer #1: Authors proposed neutrophil gelatinase-associated lipocalin partly reflects the dynamic changes of renal function among CHC patients receiving DAA. It would give another point of view for clinical daily practice. I have several questions as below.

1. In table 1, authors showed baseline characteristics of subgroup with or without grade 2/3 renal function deterioration at P12. I was confused that the P-value here (in this table) was calculated by the Student's t-test and the Mann-Whitney U test for continuous variables/the chi-square test or Fisher's exact test for categorical data to check baseline difference between subgroups; "Or" by univariate logistic regression analysis? I suggest that authors should show another table for data of univariate logistic regression analysis for each predictive factors and then recheck multivariate logistic regression analysis.

Response:

1. As we mentioned in the method, the P value in Table 1 was calculated by using the Student’s t-test and the Mann-Whitney U test for continuous variables and the chi-square test or Fisher’s exact test for categorical data, as appropriate. 

2. In order to avoid the confusion about the statistical method, we decided to use the univariate logistic regression analysis according to your suggestion (Table 1, Page 11-13, Line 194-196). We also revised the method (Page 10, Line 168-171). Abstract was also revised (Page 3, Line 42-46).

3. The final multivariate logistic regession showed the same results (Table 2, Page 15, Line 224-230). The predictive factors were age, gender, ACEI/ARB use and BL NGAL. 

2. In table 1, because many patients usually has abnormal liver function before treatment, FIB-4 might be influenced by hepatitis. I suggest "advanced liver fibrosis" might be replaced by "FIB-4 score higher than 3.25".

Response:

Thank you for your comment. 

In this study, the advanced liver fibrosis was diagnosed by an FIB-4 score higher than 3.25 or radiologic cirrhosis. The cutoff value of 3.25 was based on the metaanlysis study and total 2,297 HCV-positive paitinets regarding the liver function were included. (Chou R, Wasson N. Blood tests to diagnose fibrosis or cirrhosis in patients with chronic hepatitis C virus infection: a systematic review. Ann Intern Med 2013;158:807–820.) The sensertivity was 55% and sepecificity was 92%. Besides, according to EASL recommendations on treatment of hepatitis C, FIB-4 is generally available, simple and inexpensive, and reliable non-invasive method to assess liver disease severity prior to therapy. [J Hepatol. 2020;73(5):1170-218]. We marked the definition of advanced liver fibrosis in footnote of the Table 1 (Page 13-14, Line 203-206). 

3. In table 1, HCC history means curative or inactive or stable status of HCC. Maybe authors exclude these patients to avoid some confounding effects.

Response:

1. Thank you for your comment. We tried to exclude patients with HCC history (n=37, 16% ) and analyzed their variables. After univariate logistic regression and multivariate analyses, baseline NGAL (OR: 1.054; CI: 1.017–1.092; P= 0.004) was still predictive factor for grade 2/3 renal funtion deterioration. 

2. In the logistic regression analysis, the more variables we put in our model, the greater the sample size must be. The sample size was markedly decreased after we exlcued those paitnets with HCC history (n=37, 16%) and it casues some factors become nonsignificant. Hence, we wanted to maintain original analyses and keep HCC history as one of variables.

4. In table 2, the odds ratio (OR) of ACEI/ARB was 2.476 (P=0.048) and the OR of BL NGAL was just 1.033 (P=0.045). ACEI/ARB had the higher weight for grade 2/3 renal function deterioration. If possible, authors could exclude patients using the medications associated with renal function (in table 1, ACEI/ARB, diuretics, NSAID) to clarify the effect of BL NGAL.

Response:

1. It’s another good point to remove the effect of nephrotoxic medications by excluding these variables from original analysis. After we excluded patients taking nephrotoxic agents (n= 63, 27%), multivariate analyses revealed age (OR: 1.049; CI: 1.012–1.087; P= 0.010) and gender (OR: 2.363; CI: 1.144–4.877; P= 0.020) were predictive factors for grade 2/3 renal funtion deterioration. The decrease of sample size cause the baseline NGAL become nonsignificant factor. 

2. Subgroup analysis may be interesting if the study included more patient number. If the number is limited, it seems reasonable to adjust those factors in multivariate analyses. As previous reason, we thought the role of baseline NGAL may be better understood under larger sample size. Furthermore, to have these variables (ACEI/ARB, diuretics, NSAID) in our investigation may be helpful in realizing the roles of thses nephrotoxic medications in the renal function of CHC patients using DAA therapy. Thank you very much.

5. In Figure 3 & 4, the dynamic change of NGAL might be negatively associated with the dynamic change of eGFR from BL to P12, for patients with eGFR rank 1 and 2. This phenomenon was also found while focusing on SOF user. Therefore, these findings demonstrated the dynamic change of NGAL might reflect the dynamic changes of renal function. Maybe, authors could add the dynamic change of NGAL as one of predictive factors for analysis. As for BL NGAL, it should recheck by 

univariate/multivariate logistic regression analysis.

Response:

Thank you for your reminder. 

1. Althrough the decrease of eGFR was noted in paitnets with eGFR rank 1 and 2, the serum level of NGAL showed no significant differences from BL to P12 (Figure 4). The dynamic changes of NGAL were not negatively associated with the dynamic change of eGFR from BL to P12 for the patients with baseline rank 1 and 2 renal function. We think the values of eGFR and NGAL were not clearly demonstrated under previous versions of figure due to smaller font size. Hence, we provided newer figures in order to improve the legibility. 

2. For the SOF user, the dynamic change of NGAL might be negatively associated with the dynamic change of eGFR from BL to P12 (Figure 3). But the associations between eGFR and NGAL changes were not statistically significant and they were shown in the following table.

 P12-BL EOT-BL P12-EOT

eGFR (mean±SD) -2.02±10.44 -2.52±9.87 0.49±9.33

NGAL (mean±SD) 0.73±2.99 0.29±2.47 0.44±3.36

Linear correlation R = -0.144 , P =0.131 R = 0.007 , P =0.939 R = -0.052 , P =0.585

3. As to the dynamic change of NGAL as possible predictive factor, we tried to analyze the difference between NGAL (EOT) and NGAL (BL) as a variable for those with or without grade 2/3 renal function deterioration in the following table. We found the dynamic change between EOT and BL or P12 and BL would not become a significant predictive factor after univariate logistic regression analyses. Hence, it seemed that the dynamic change of NGAL plays no role in predicting the renal function change.

Variable All patients 

(n = 232) With grade 2/3 deterioration

(n = 62) (26.7%) Without grade 2/3 deterioration 

(n = 170) (73.3%) P-value#

Difference between NGAL (EOT) and NGAL (BL) 0.04±2.91 -0.26±2.37 0.14±3.08 0.346

Difference between NGAL (P12) and NGAL (BL) 0.39±3.41 0.29±3.74 0.43±3.29 0.772

Reviewer #2: This retrospective study was aimed to investigate the changes in eGFR and the role of neutrophil gelatinase-associated lipocalin (NGAL) in CHC patients receiving DAA, including nonSOF- and SOF-based regimens. Although it is interesting, some concerns are needed to be clarified.

1. Since SOF might be related to renal injury via drug-induced tubulointerstitial nephritis, the proportion of SOF-based DAA users should be added in Table 1.

Response:

The proportion of nonSOF-based and SOF-based users were shown in table 1 (Page 12, table 1, the variable “Sofosbuvir-based”). There was a total of 112 (48.3%) SOF users. Of these patients, 33 (53.2%) were in grade 2/3 renal function deterioration group and 79 (46.5%) were in the group without grade 2/3 deterioration (P=0.363). Thank you very much.

2. Factors associated with grade 2/3 renal function deterioration in CHC patients receiving DAA at P12 in nonSOF-based and SOF-based DAA users may be analyzed in another two tables, respectively. The cut -off of variables, including age, BL NGAL, should be offered in Table 2.

Response:

1. Factors associated with grade 2/3 renal function deterioration were different in nonSOF and SOF-based users. Therefore, we showed the new tables in supplement materials to demonstrate the difference between nonSOF (n= 120) and SOF-based (n=112) DAA users. The predictive factors of nonSOF users (n= 120) were gender (OR:3.161; CI: 1.150–8.686; P=0.026), fatty liver (OR: 4.684; CI: 1.689–12.990; P=0.003), and hyperlipidemia (OR:9.401; CI:1.106-7.333; P=0.002). The basline NGAL has the trend associated with the grade 2/3 renal function deterioration (P=0.05). The ACEI/ARB users (OR:3.276; CI: 1.008–10.647;P=0.049) was the only predective factor for SOF users (n=112). The possible mechanisms to explain different predictive factors of grade 2/3 renal function deterioration in nonSOF and SOF DAA users are unclear and require further study to clarify it. We added this in the our result (Page 16-17, Line 257-265) and discussion (Page 21-22, Line 356-360). These tables are displayed as supplementary materials. 

2. We analyzed age and BL NGAL as continuous variables in the multivariate logistic regession. Therefore, we don’t show the cut-off values in table 2. 

Supplement table 1. Baseline characteristics of chronic hepatitis C patients receiving DAA with or without grade 2/3 renal function deterioration at P12 in nonSOF users.

Variable All patients 

(n = 120) With grade 2/3 deterioration

(n = 29) (26.7%) Without grade 2/3 deterioration 

(n = 91) (73.3%) P value

Baseline clinical characteristics

Age (years)† 64.06 ± 10.22 67.15 ± 6.30 63.10 ± 11.03 0.043

Male (%) 41 (34.2%) 14 (48.3%) 27 (29.7%) 0.076

Fatty liver 38 (31.7%) 15 (51.7%) 23 (25.3%) 0.011

Hyperlipidemia 12 (10.0%) 7 (24.1%) 5 (5.5%) 0.008

Diabetes mellitus 26 (21.7%) 9 (31.0%) 17 (18.7%) 0.196

Hypertension 26 (21.7%) 9 (31.0%) 17 (18.7%) 0.196

eGFR ranks* 0.428

 rank 1 38 (31.7%) 7 (24.1%) 31 (34.1%) 

 rank 2 70 (58.3%) 19 (65.5%) 51 (56.0%) 

 rank 3 12 (10.0%) 3 (10.3%) 9 (9.9%) 

Baseline characteristics of HCV and liver-related conditions

Advanced fibrosis (%) 70 (58.3%) 19 (65.5%) 51 (56.0%) 0.396

HCC history (%) 20 (16.7%) 6 (20.7%) 14 (15.4%) 0.569

Splenomegaly (%) 37 (30.8%) 13 (44.8%) 24 (26.4%) 0.069

Ascites (%) 1 (0.8%) 0 (0.0%) 1 (1.1 %) 1.000

Baseline HCV viral load (IU/mL)† 6.10Log ± 0.86Log 5.86Log ± 1.20Log 6.17Log ± 0.71Log 0.512

HCV genotype 1 (%) 110 (91.7%) 26 (89.7%) 84 (92.3%) 0.703

Baseline medications associated with renal function

ACEI/ARB users 12 (10.0%) 4 (13.8%) 8 (8.8%) 0.481

Diuretics users 2 (1.7%) 0 (0.0%) 2 (2.2%) 1.000

NSAID users 21 (17.5%) 5 (17.2%) 16 (17.6%) 1.000

Baseline laboratory data

Baseline NGAL (ng/ml) † 17.55 ± 9.55 20.59 ± 10.32 16.58 ± 9.14 0.042

ALT(U/L)† 83.63 ± 72.30 99.21 ± 99.57 78.66 ± 61.02 0.498

AST(U/L)† 60.55 ± 62.40 73.79 ± 86.03 56.33 ± 52.65 0.406

Albumin (g/dl) † 4.27 ± 0.30 4.22 ± 0.29 4.29 ± 0.30 0.222

Total bilirubin (mg/dl) † 0.73 ± 0.30 0.78 ± 0.32 0.71 ± 0.29 0.399

eGFR (ml/min/1.73m2)† 80.66 ± 15.93 78.72 ± 13.99 81.28 ± 16.52 0.274

Hb† (gm/dL) 13.67 ± 1.48 13.71 ± 1.54 13.65 ± 1.46 0.688

Prothrombin time (INR)† 1.02 ± 0.05 1.04 ± 0.06 1.01 ± 0.05 0.078

e-GFR, estimated glomerular filtration rate; HCC, hepatocellular carcinoma; HCV, hepatitis C virus; ACEI, angiotensin-converting enzyme inhibitor; ARB, angiotensin receptor blocker; NSAID, nonsteroidal anti-inflammatory drugs; NGAL, neutrophil gelatinase-associated lipocalin; ALT, alanine aminotransferase; AST, aspartate aminotransferase; INR, international normalized ratio

*rank 1: > 90 ml/min/1.73 m2, rank 2: 60-90 ml/min/1.73 m2, rank 3: 30-60 ml/min/1.73 m2; †Data are expressed as mean±SD.

Supplement table 2. Baseline characteristics of chronic hepatitis C patients receiving DAA with or without grade 2/3 renal function deterioration at P12 for SOF users.

Variable All patients 

(n = 112) With grade 2/3 deterioration

(n = 33) (26.7%) Without grade 2/3 deterioration 

(n = 79) (73.3%) P value

Baseline clinical characteristics

Age (years)† 63.98 ± 11.14 65.82 ± 10.09 63.21 ± 11.52 0.255

Male (%) 42 (37.5%) 14 (42.4%) 28 (35.4%) 0.525

Fatty liver 37 (33.0%) 8 (24.2%) 29 (36.7%) 0.271

Hyperlipidemia 6 (5.4%) 1 (3.0%) 5 (6.3%) 0.668

Diabetes mellitus 18 (16.1%) 5 (15.2%) 13 (16.5%) 1.000

Hypertension 24 (21.4%) 7 (21.2%) 17 (21.5%) 1.000

eGFR ranks* 0.382

 rank 1 38 (33.9%) 10 (30.3%) 28 (35.4%) 

 rank 2 50 (44.6%) 14 (42.4%) 36 (45.6%) 

 rank 3 24 (21.4%) 9 (27.3%) 15 (19.0%) 

Baseline characteristics of HCV and liver-related conditions

Advanced fibrosis (%) 83 (74.1%) 26 (78.8%) 57 (72.2%) 0.637

HCC history (%) 17 (15.2%) 8 (24.2%) 9 (11.4%) 0.146

Splenomegaly (%) 34 (30.4%) 12 (36.4%) 22 (27.8%) 0.377

Ascites (%) 4 (3.6%) 2 (6.1%) 2 (2.5 %) 0.580

Baseline HCV viral load (IU/mL)† 5.88Log ± 1.08Log 5.87Log ± 1.10Log 5.88Log ± 1.08Log 0.924

HCV genotype 1 (%) 44 (39.3%) 17 (51.5%) 27 (34.2%) 0.095

Baseline medications associated with renal function

ACEI/ARB users 13 (11.6%) 7 (21.2%) 6 (7.6%) 0.054

Diuretics users 9 (8.0%) 4 (12.1%) 5 (6.3%) 0.445

NSAID users 13 (11.6%) 2 (6.1%) 11 (13.9%) 0.339

Baseline laboratory data

Baseline NGAL (ng/ml) † 14.54 ± 8.20 15.73 ± 9.27 14.05 ± 7.72 0.612

ALT(U/L)† 89.80 ± 78.38 87.70 ± 63.70 90.67 ± 84.11 0.532

AST(U/L)† 62.93 ± 46.57 66.06 ± 45.85 61.62 ± 47.10 0.491

Albumin (g/dl) † 4.12 ± 0.41 4.07 ± 0.45 4.14 ± 0.39 0.650

Total bilirubin (mg/dl) † 0.85 ± 0.48 0.87 ± 0.44 0.84 ± 0.50 0.559

eGFR (ml/min/1.73m2)† 77.10 ± 19.54 74.41 ± 19.31 78.22 ± 19.65 0.330

Hb† (gm/dL) 13.32 ± 1.75 13.28 ± 1.52 13.34 ± 1.85 0.959

Prothrombin time (INR)† 1.04 ± 0.08 1.05 ± 0.08 1.04 ± 0.08 0.529

e-GFR, estimated glomerular filtration rate; HCC, hepatocellular carcinoma; HCV, hepatitis C virus; ACEI, angiotensin-converting enzyme inhibitor; ARB, angiotensin receptor blocker; NSAID, nonsteroidal anti-inflammatory drugs; NGAL, neutrophil gelatinase-associated lipocalin; ALT, alanine aminotransferase; AST, aspartate aminotransferase; INR, international normalized ratio

*rank 1: > 90 ml/min/1.73 m2, rank 2: 60-90 ml/min/1.73 m2, rank 3: 30-60 ml/min/1.73 m2; †Data are expressed as mean±SD.

Supplement table 3. Patient factors associated with grade 2/3 renal function deterioration in chronic hepatitis C patients receiving DAA at P12^ for nonSOF and SOF-based DAA users.

 Odds Ratio (95% CI) P value

NonSOF-based DAA users

Sex

 Female

 Male 

1.000

3.161 (1.150–8.686) 

0.026

Fatty liver

 No

 Yes 

1.000

4.684 (1.689–12.990) 

0.003

Hyperlipidemia

 No

 Yes 

1.000

9.401 (2.268–38.959) 

0.002

BL NGAL 1.051 (1.000–1.104) 0.05

SOF-based DAA users

ACEI/ARB users

 No

 Yes 

1.000

3.276 (1.008–10.647) 

0.049

^Adjusted for age, sex, variables with P < 0.1 from supplement table 1: fatty liver, hyperlipidemia, splenomegaly, BL NGAL and PT INR and from supplement table 2: GT1 and ACEI/ARB users for nonSOF users and SOF users, respectively ; factors reported to be associated with renal injury: baseline renal disease, DM, HTN, liver disease (advanced fibrosis), ACEI/ARB users, diuretics users, and NSAID users were also considered in multivariate analysis.

3. Page 16, line 3, “The levels of NGAL at the BL, EOT, and P12 were all significantly different between nonSOF- and SOF-based DAA users (P = 0.011, 0.024, and 0.037, respectively). Please add a figure.

Response:

Thank you for your commnet. The figure has been revealed as figure 3b. We also provided new figures for better legibility and added the footnote after this sentence (Page 16, line 257).

4. Page 21, line 6, “Although we found that an increase in NGAL was significant at P12 by SOF-based DAA, the values were still within the normal range”. What is the normal range of NGAL?

Response:

Thank you for your reminder. We showed the normal range of NGAL in the page 9, method paragraph 2.4 and line 163-164. We added this normal range again in Page 21, line 356 for better readability.

5. Page 22, “Fig 4. The eGFR and NGAL changes from the BL, EOT to P12 in different BL eGFR ranks in CHC patients receiving DAA therapy. a. eGFR; b. NGAL” should be in “3. Results”.

Response:

 Thank you for your reminder. We have corrected it (Page 18, line 286-287). Thank you very much.

---

## [Decision Letter · Decision Letter 1]

9 Aug 2021

Neutrophil gelatinase-associated lipocalin partly reflects the dynamic changes of renal function among chronic hepatitis C patients receiving direct-acting antivirals

PONE-D-21-15486R1

Dear Dr. Chen,

We’re pleased to inform you that your manuscript has been judged scientifically suitable for publication and will be formally accepted for publication once it meets all outstanding technical requirements.

Kind regards,

Chen-Hua Liu

Academic Editor

PLOS ONE

Reviewers' comments:

Reviewer's Responses to Questions

**Comments to the Author**

1. If the authors have adequately addressed your comments raised in a previous round of review and you feel that this manuscript is now acceptable for publication, you may indicate that here to bypass the “Comments to the Author” section, enter your conflict of interest statement in the “Confidential to Editor” section, and submit your "Accept" recommendation.

Reviewer #1: All comments have been addressed

Reviewer #2: All comments have been addressed

2. Is the manuscript technically sound, and do the data support the conclusions?

Reviewer #1: Yes

Reviewer #2: Yes

3. Has the statistical analysis been performed appropriately and rigorously? 

Reviewer #1: Yes

Reviewer #2: Yes

4. Have the authors made all data underlying the findings in their manuscript fully available?

Reviewer #1: Yes

Reviewer #2: Yes

5. Is the manuscript presented in an intelligible fashion and written in standard English?

Reviewer #1: Yes

Reviewer #2: Yes

6. Review Comments to the Author

Reviewer #1: The authors answered and revised the manuscript according to my suggestions. I endorse publication of the manuscript.

Reviewer #2: (No Response)

7. PLOS authors have the option to publish the peer review history of their article (what does this mean?). If published, this will include your full peer review and any attached files.

Reviewer #1: No

Reviewer #2: **Yes: **Wei-Yu Kao

---

## [Editor Report · Acceptance letter]

18 Aug 2021

PONE-D-21-15486R1 

Neutrophil gelatinase-associated lipocalin partly reflects the dynamic changes of renal function among chronic hepatitis C patients receiving direct-acting antivirals 

Dear Dr. Chen:

I'm pleased to inform you that your manuscript has been deemed suitable for publication in PLOS ONE. Congratulations! Your manuscript is now with our production department. 

Kind regards, 

on behalf of

Dr. Chen-Hua Liu 

Academic Editor

PLOS ONE